# Properties of wood composite plastics made from predominant Low Density Polyethylene (LDPE) plastics and their degradability in nature

Arif Nuryawan[1¶]*, Nova O. Hutauruk[1☯], Esra Yunita S. Purba[1☯], Nanang Masruchin[2‡], Ridwanti Batubara[1☯¶], Iwan Risnasari[1‡¶], Fatih Khusno Satrio[3], Rahmawaty[4‡], Mohammad Basyuni[5¶], Deirdre McKay[6]

1 Department of Forest Products Technology, Faculty of Forestry, Universitas Sumatera Utara, Medan, North Sumatra, Indonesia, 2 Research Center for Biomaterials, Indonesian Institute of Sciences (LIPI), Cibinong, West Java, Indonesia, 3 PT Horiba Indonesia, Tangerang, Banten, Indonesia, 4 Department of Forest Management, Faculty of Forestry, Universitas Sumatera Utara, Medan, North Sumatra, Indonesia, 5 Department of Silviculture, Faculty of Forestry, Universitas Sumatera Utara, Medan, North Sumatra, Indonesia, 6 School of Geography, Geology and the Environment, Keele University, Keele, United Kingdom

☯ These authors contributed equally to this work.
‡ These authors also contributed equally to this work.
¶ Membership of the Center Excellence for Mangrove, Universitas Sumatera Utara, Medan, North Sumatra, Indonesia.
* arif5@usu.ac.id

**Data Availability Statement:** All relevant data are within the manuscript and its Supporting Information files.

## Abstract

To address concerns over plastics in the global environment, this project produced three wood plastics composites (WPCs) which could divert plastics from the waste stream into new materials. The three materials made had a ratio of 85%:15%, 90%:10%, and 95%:5% low density polyethylene (LDPE) to wood powder and were produced using the dissolution method. Physical and mechanical properties of each WPC were evaluated according to Japanese Industrial Standard (JIS) A 5908:2003. Their degradation in nature was evaluated through a graveyard test and assay test conducted in *Coptotermes curvignathus* termites. Results showed that density, moisture content, thickness swelling and water absorption of the WPCs fulfilled the JIS standard. The mechanical properties of these composites also met the JIS standard, particularly their modulus of elasticity (MOE). Modulus of rupture (MOR) and internal bonding (IB) showed in lower values, depending on the proportion of wood filler they contained. Discoloration of the WPCs was observed after burial in the soil with spectra alteration of attenuated transmission reflectance (ATR) in the band of 500–1000 cm$^{-1}$ which could be assigned to detach the interphase between wood and plastics. As termite bait, the WPCs decreased in weight, even though the mass loss was comparatively small. Micro Confocal Raman Imaging Spectrometer revealed that termite guts from insects feeding on WPCs contained small amounts of LDPE. This indicated termite can consume plastics in the form of WPCs. Thus WPCs made predominantly of plastics can be degraded in nature. While producing WPCs can assist in decreasing plastics litter in the environment, the eventual fate of the LDPE in termites is still unknown.

**Funding:** AN thanks to Universitas Sumatera Utara (USU) for funding this research under scheme of Penelitian Terapan TALENTA-USU year of 2018, contract number 2590/UN5.1.R/PPM/2018 date of March 16, 2018. FKS is employed by and receives salary from PT.Horiba Indonesia. The funders had no role in study design, data collection and analysis, decision to publish, or preparation of the manuscript.

**Competing interests:** FKS is employed by and receives salary from PT.Horiba Indonesia. This does not alter our adherence to PLOS ONE policies on sharing data and materials.

## Introduction

Production and consumption of plastics worldwide has increased rapidly since the 1950s [1]. Plastics are now ubiquitous, used to deliver food (e.g. as packaging, food containers, and beverage bottles), produce textiles and synthetic fibers (e.g. as polyester cloth and rope), and as building materials (e.g. electrical insulation, pipes, and window frames). As materials, plastics offer ease of processing, good ductility, high toughness, excellent chemical resistance [2] and moldability. However, plastics durability is problematic; these materials are very difficult to degrade. Plastic pollution in the natural environment is now a global problem and solutions require efforts to reduce the amount of plastics going to waste.

Attempts to repair, recondition, remanufacture, and recycle plastics still produce plastic debris [3]. Materials disposed to municipal solid waste require a very long time to deteriorate. Some scientists have been exploring new types of plastics, for example biopolymers [4] and bio-degradable plastics [5], or mixtures of bio-plastics with natural fillers [6,7] in order to substitute the current suite of plastic materials with materials that will potentially cause less environmental harm. However, these efforts remain fairly localized, depending the local government policies and waste management strategies within each nation [8].

The potential in combining hydrophobic plastics and hydrophilic natural resources such as lignocelluloses materials has attracted increasing attentions over the past three decades [9]. These mixtures, which consist of a plastics matrix and lignocelluloses filler, are wood plastics composites (WPCs). WPCs have desirable properties, improved mechanical strengths and other working characteristics that enable them to substitute for traditional materials such as metal and wood [10] and, indeed, plastic itself.

Numerous studies of WPCs have explored their formulation, the choice of plastics matrix and natural filler content, the compatibility between the two, and examined the coupling agents because both these materials have very distinct characteristics. For example, Nygard et al. [11] processed WPCs with various grades of polypropylene (PP) plastics as the matrix polymer and wood powder and wood chips as filler, using a 50:50 ratio. Well-dispersed wood powders in the extruded compounds and injection molded test samples were shown in Scanning Electron Microscope (SEM) examination. Further, the dispersion of the wood fiber was significantly improved by introduction of compatibilizers, such as maleic anhydride, cellulose esters, etc. Similar work presented by Rao et al. [12] made WPCs using recycle polyethylene (rPE) up to 40%with different coupling agents. Rao's team used fluorescence microscopy and SEM to evaluatethe chemical structure of bonding and the micro-morphological features of the WPCs. Our group [13] also used SEM to assess bio-composites made of high density polyethylene (HDPE) and wood flour with and without a coupling agent. We found the role of the coupling agent was important in surface adhesion, particularly in the interface region. In other work, Essabir et al. [14] incorporated PP and up to 30% filler (nuts shell particles) in a bio-composite with a styrene-(ethylene-butene)-styrene triblock copolymer grafted with maleic anhydride (SEB-g-MA) as a coupling agent using extrusion/co-extrusion methods. In their study, the filler did not significantly modify the thermal stability of PP but the use of a coupling agent did. Jam & Behravesh [15] used wood particle filler to produce a WPC via injection molding. Their findings suggest that the processing of composites containing wood content above 60% was highly challenging, particularly where the wood particles were fine type. Thus research to date suggests WPCs can be made of a plastics and filler mixture with a ratio of up to 30% plastics: 70% wood using a process of mixing between plastics and filler consisting of compounding, compression, extrusion, co-extrusion and injection methods, with or without using coupling agent. A comprehensive literature review of the traditional method of mixing between plastics and filler-solvent extraction or dissolution- by Zhao et al. [16] concurs. They

identified these methods as having potential to recycling mass-produced plastics into environmentally more benign and potentially profitable materials.

Our research produced WPCs by simple solvent extraction with low-density polyethylene (LDPE) plastics as the matrix and only small amount of wood filler. We choose the highest plastic ratio feasible to maximize the potential benefits of rerouting plastic waste. We selected LDPE because it is one of the major types of polyolefin thermoplastics used worldwide in applications such as bags, toys, containers, pipes, etc [17] and therefore widely available. To understand the potential degradation WPCs made predominantly of plastics in nature, we undertook studies of its deterioration. Two reviews of the literature on biological degradation of WPCs [18,19] observe that WPCs may be attacked by microorganisms. While there are standards available to evaluate the bio-deterioration of WPCs, however there are no available reports on termite attack. In the literature, bio-degradation of WPCs was caused by decay fungi, moulds, stain fungi, and marine borers. Even though there were studies of the bio-physic-degradation process of WPCs such as their mold resistance [20], fungi [21] and weathering [10], these studies discussed WPC made of bio-plastics. Therefore, there is an information gap as to whether or not a WPC made of predominantly non biodegradable plastics may be consumed by termites.

The objectives of this study were to evaluate the physical and mechanical properties of WPCs manufactured from LDPE and small amount of wood filler using dissolution/ re-precipitation technique and to explore their potential degradation in the natural environment through a termite assay.

## Materials and methods

### Materials

The materials used in this study were commercial granule virgin thermoplastic LDPE and wood powder or wood flour (WF). The LDPE had a density of 0.93 g/cm$^3$ and melting point of 110˚C and was used as the matrix. The WF used was originated from the sawdust waste of durian-wood (*Durio zibethinus*). Prior to mixing with LDPE, the WF was sieved using 80 mesh screen and then dried in a convection oven for 24 h at (103±2)˚C. To aid in dissolution process, xylene (reagent grade) was used as the single solvent. Selection of xylene as the solvent was based on work of Il'yasova et al. [22] which demonstrates that LDPE can dissolve in xylene homogenously even though Hilderbrand solubility parameter (δ) of the xylene was the lowest among toluene, trichloroethylene, chlorobenzene, and benzene [23]. Hadi et al. [24] further observed that the solubility of LDPE in pure solvent was very good compared to a blended solvent, making xylene the best choice.

### Dissolution/Re-precipitation technique

The experimental procedure was comprised of three stages: dissolution of the plastics, addition of the WF, and evaporation of the solvent. Dissolution of LDPE used glass reactor equipped with a stirrer and mantle heater. LDPE and xylene with a ratio 1:20 (w/v) were placed in the reactor. The system was heated up to 115˚C to allow the plastics to melt and dissolve. After approximately 30 minutes, when all the plastics were dissolved, WF was introduced at varying ratios (by weight) of 95:5; 90:10; and 85:15 of plastics and WF, respectively. The blend was then stirred gently for homogenization approximately for an hour. Following this, the mixture was transformed into pellets and placed in the acid chamber overnight to allow the xylene to evaporate.

## Composite processing

All the WPCs were manufactured in a two-step pressing process: hot pressing and cold pressing, respectively, with a target density of 0.70 g/cm³. The pellets produced were first placed into a prepared steel-mould with dimension of 25 cm x 25 cm x 0.5 cm. Then, the hot flat-platen pressing process was applied using temperature of 115°C, pressure of 30 kgf/cm², and a duration of 6 minutes to allow the pellets to melt. After exposure to the hot-press, the WPC board was carefully taken out cold pressing. Cold pressing was carried out using steel loading with a weight of 15 kg for 24 hours in ambient temperature (27°C) in order to transform the WPC into a solid product.

## Test methods

**Physical properties of the WPC.** Physical properties of the resulting WPC investigated included density, moisture content, and dimensional stability. The first two were measured gravimetrically in accordance with JIS A 5908 [25] using WPC specimens with dimensions of 10 cm x 10 cmx 0.5 cm. Prior to being oven-dried at 105°C for 24 h to obtain the oven-dry weight ($m_0$), all of the WPC specimens were weighed ($m_1$) and measured. The mean dimensions of length, width, and thickness of the test piece with which the volume was calculated in order to determine the density was calculated. Moisture content (MC) was calculated using Eq (1):

$$MC(\%) = \frac{(m_1 - m_0)}{m_0} x\ 100 \tag{1}$$

Dimensional stability was shown through both water absorption (WA) and thickness swelling (TS) tests. Specimens of a size of 5 cm x 5 cm x 0.5 cmwere immersed in water for 2 and 24 hours at ambient temperature (27°C). After wiping off the water, the weight gain was determined by the weight before ($w_1$) and after the immersion test ($w_2$) using the formula (2). TS was then determined by measuring the thickness before ($t_1$) and after ($t_2$) immersion in the water using formula (3).

$$WA(\%) = \frac{(w_2 - w_1)}{w_1} x\ 100 \tag{2}$$

$$TS(\%) = \frac{(t_2 - t_1)}{t_1} x\ 100 \tag{3}$$

Statistical analyses were conducted on all the physical properties parameters by applying the analysis of variance (ANOVA) followed by Duncan Multi Range Test (DMRT) for comparison of all treatments. Differences were considered significant at $P < 0.05$.

**Mechanical testing of the WPC.** Measurements of the maximum load ($P$) as well as loads before proportional limits ($\Delta P$) and its deflection ($\Delta y$) were carried out using test apparatus of a one point loading of Tensilon Universal Testing Machine (UTM) at a crosshead speed of 10 mm/ min with horizontal position, sample size of 200 mm x 50 mm x 5 mm and three replications for each condition according to JIS A 5908 [25]. Bending strength or modulus of rupture (MOR) was calculated using Eq (4), where span (L), width of test piece (b), and thickness of test piece (t) were determined in mm. Further, bending Young's modulus or modulus of elasticity (MOE) was calculated using Eq (5).

$$MOR(^N/_{mm^2}) = \frac{3PL}{2bt^2} \tag{4}$$

$$MOE\left(^{N}/_{mm^2}\right) = \frac{\Delta PL^3}{4bt^3\Delta y} \tag{5}$$

In order to evaluate the bonding strength within the WPC, an internal bond (IB) test with specimen size of 5 cm x 5 cm x 0.5 cm and three replications for each condition was also conducted using UTM. Maximum load ($P'$) at the time of failing force or breaking load of perpendicular tensile strength of the WPC was calculated using formula (6). In this test, the tension loading speed was 2 mm/min.

$$IB\left(^{N}/_{mm^2}\right) = \frac{P'}{2bL} \tag{6}$$

Statistical analyses were conducted on all the mechanical properties parameters by applying the ANOVA and DMRT.

**Degradation of the WPC.** The graveyard test was used to examine the degradation of the WPCs. This test is relevant to the fate of waste plastics buried in municipal waste or soil. The test was carried out by burying the WPC samples and a control (durian wood) with size of 5 cm x 5 cmx 0.5 cm and three replications for each condition in a moist soil test pit outside our laboratory in Medan, North Sumatra, Indonesia with exposure condition as follows: temperature range was 30.21–31.68°C, air humidity range was 68.82–70.21%, and pH range was 6.85–7.63. The samples were observed for weight loss in every 10 days and the experiment finished after 50 days. Weight loss was recorded. Further examinations were then carried out on the WPC's durability via microscopy and infra-red (IR) spectroscopy. Weight loss ($WL$) was determined by oven-dry weight sample before ($w_1$) and after ($w_2$) the WPC samples were buried in the soil using formula (7).

$$WL(\%) = \frac{(w_1 - w_2)}{w_2}x\ 100 \tag{7}$$

The final results obtained for this treatment initially were evaluated by ANOVA followed by Tukey test at $\alpha = 0.05$. Tukey test was performed for one-factor analysis.

Microscopy studies included color analysis were conducted using CIE Lab method as defined by CIE (*Commissions Internationale d'Eclairage*) [26]. Color comparison was conducted on WPCs before and after the 50 days graveyard test. The value of CIE_L* describes color lightness; 0 for black and L* for white. Dimension of CIE_a* describes green-red; -a* indicates green while +a* indicates red. Dimension of CIE_b* describes blue-yellow; -b* indicates blue while +b* indicates yellow. The color difference ($\Delta E$) was calculated using Eq (8) where $\Delta L^*$ was lightness difference (lighter or darker) (L* sample − L*control), $\Delta a^*$ was green or red difference (a*sample-a*control), and $\Delta b^*$ was blue-yellow difference (b*sample-b*control).

$$\Delta E = \sqrt{\Delta L^{*2} + \Delta a^{*2} + \Delta b^{*2}} \tag{8}$$

The effect of $\Delta E$ was classified according to Table 1.

Spectroscopy was included both IR analysis on the WPC samples before and after the graveyard test and thermal analysis. IR analysis employed ATR (attenuated transmission reflectance) to identify functional groups present at the WPC's surface before and after the graveyard test. For each WPC sample, 100 scans were recorded with wave number range 500–4000 cm$^{-1}$ to examine chemical compound alterations during the graveyard test. Further thermal analysis using DTA (differential thermal analysis) demonstrated the resistance of

**Table 1. Classification of the color difference between before and after test.**

| Value of $\Delta E$ | Effect |
| --- | --- |
| <0.2 | not seen |
| 0.2–1.0 | very little |
| 1.0–3.0 | little |
| 3.0–6.0 | moderate |
| >6.0 | high |

either the WPC samples or plastics to high temperature. This examination provided information on the nature of the samples, including the analysis of nonhomogeneous samples such as composites [27].

Termite digestion of the WPCs was confirmed using micro confocal Raman imaging spectrometry. This termite assay test was conducted by providing WPC samples (without durian wood as control) as the sole food source to *Coptotermes curvignathus* termites held in aquarium box filled with moist soil in room temperature for 2 (two) weeks. The termites had no choice of food other than the wood sawdust particles incorporated in the plastics matrix of the WPC samples. In this circumstance, if the termite, which typically feeds on durian wood, attacks the WPC's wood filler for food, the plastics matrix plastics should be broken up either physically, mechanically or biologically. Termites will first bite the surface and then the interior of the WPC to feed on the wood filler for feed. Physically and mechanically, the WPC will be broken down. Biologically, the termites presumably ingest some part of the plastics incidentally [28] as shown in Fig 1a.

100 termites (*C.curvignathus*) were acclimatized to the aquarium for 10 days after collection from their native habitat (from Tri Dharma Forest inside Universitas Sumatera Utara campus, as shown in Fig 1b). Observations and calculations were conducted daily to record living and dead termites for a week. Mortality rate (MR) of the termites was calculated daily using formula (9) where $N_1$ = total number of termite (100 termites) and $N_2$ = total number of dead

(a)                                                             (b)

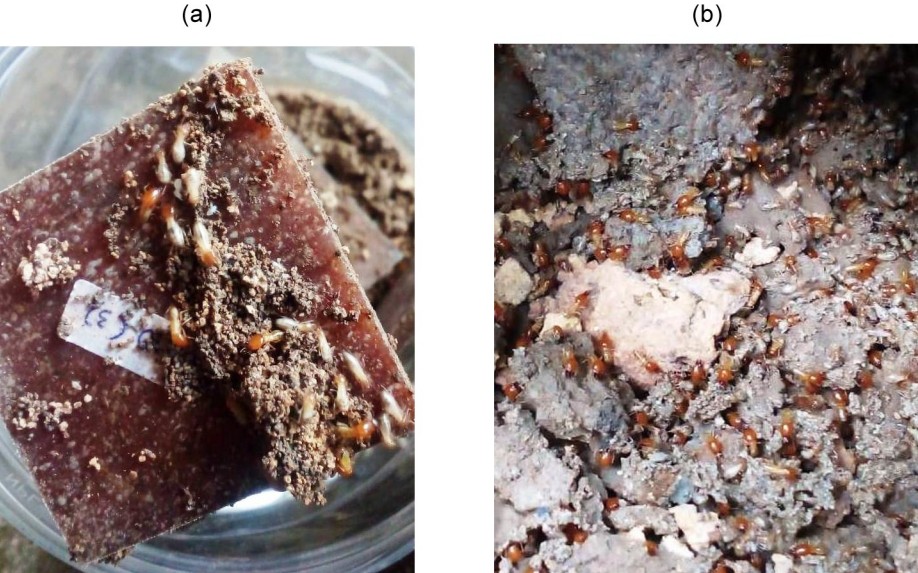

**Fig 1. *C.curvignathus* termites (a) with WPCs as food source (b) in native habitat.**

termites.

$$MR(\%) = \frac{N_2}{N_1} \text{x } 100 \tag{9}$$

Weight loss (WL) for the WPC was determined by oven-dry weight sample before ($w_1$) and after ($w_2$) termite assay using formula (10).

$$WL(\%) = \frac{(w_1 - w_2)}{w_2} \text{x } 100 \tag{10}$$

The resistance of the WPCs to termites was classified according to Indonesian Standard (SNI 01–7207) [29] in Table 2.

Further, the results obtained for this treatment initially were evaluated by ANOVA followed by Tukey test at $\alpha = 0.05$ to verify whether difference between means was statistically significant.

In order to confirm whether termitesingest the plastics within the WPCs, a sample of termites was prepared for examination by termite surgery. Following Antriana [30], the termite's thorax was clamped and the abdomen stabbed using a syringe, pushing the insect spinally onto a slide glass. A light microscope was used to locate the termite's guts in preparation for Raman spectroscopy. In this procedure, a micro confocal Raman imaging spectrometer (Horiba Scientific Lab RAM HR evolution) with a spectra range of 100 to 4000 cm$^{-1}$ equipped with a grating (600 grooves/mm; 750 nm blazed angle) and a laser operating at a wavelength of 785 nm coupled with a Raman filter of 785 edge filter (Stokes Raman) was employed to record if the spectra of the observation object—the termite guts—fit in the LDPE spectra range. LDPE has a spectra range of 2700 to 3200 cm$^{-1}$. The Raman spectra were recorded in a hole of 1000 μm and a 50-fold magnifications objective of NIR (Nir Infra-Red).

## Results and discussions

### Physical properties of the WPCs produced

The density and moisture content of the WPCs produced are shown in Figs 2 and 3.

The densities of WPCs with varying amount of wood filler are shown in Fig 2. Even though the density values met the standard [25] which required 0.40–0.90 g/cm$^3$, they were still below target (0.70 g/cm$^3$). This discrepancy is mainly due to the agglomeration of the wood filler as the wood's natural lumens. Both lead to voids forming inside the WPC system, thus increasing the thickness of the WPC. Consequently, volume of the WPC specimens increase, decrease its density and further influencing its mechanical properties. However, addition of WF up to 15% in the WPC system showed statistically non significant.

**Table 2. Classification of the resistance class to termite assay.**

| Class | Weight loss (%) | Class of resistance |
|-------|-----------------|---------------------|
| I     | < 3.52          | Very durable        |
| II    | 3.52–7.50       | Durable             |
| III   | 7.50–10.96      | Moderate            |
| IV    | 10.96–18.94     | Poor                |
| V     | 18.94–31.89     | Very poor           |

Source: [29]

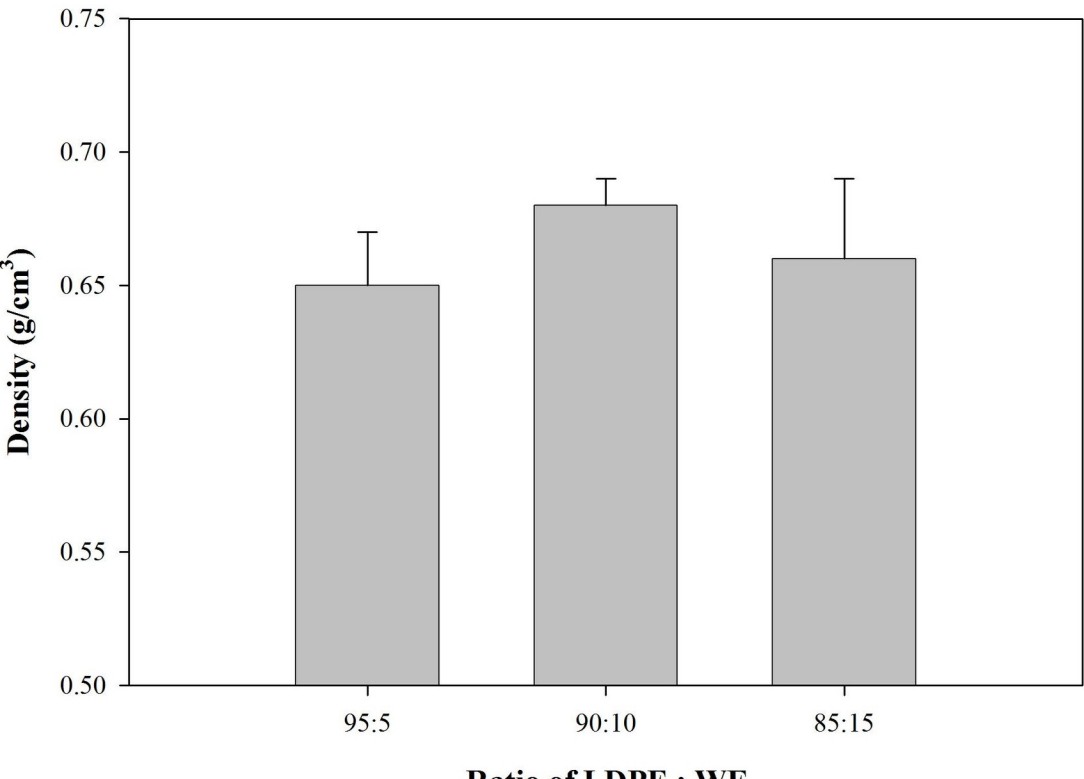

**Fig 2. Density of predominantly plastic WPCs.**

Moisture content of the WPC is presented in Fig 3. The values for moisture content were very low because hydrophobic plastics formed the major part of the composite. However, the moisture content of the WPC with the highest proportion of wood filler content was significantly different statistically. This suggests that wood filler determined the final moisture content of the WPC. The literature offers no specific reports on initial moisture content of WPCs except for discussions of water uptake and durability of the WPCs [31–33].

Dimensional stability of the WPCs was shown both in thickness swelling and water absorption tests for 2 and 24 hours, depicted in Figs 4 and 5 respectively. At the initial stage (2 h), thickness swelling as well as water sorption fluctuated. In the final stage (24 h) both properties increased proportionally with the amount of wood filler present. There was a statistically significant change in the test specimen both dimensions of thickness and water absorption with high wood filler content. Water sorption in WPCs is an important quality indicator because such composites absorb less moisture and do so more slowly than solid wood [34]. WPCs were also more resistant to fungal decay and possessed better dimensional stability when exposed to moisture [9,35]. Water absorption capacity is affected by the nature of the wood filler and the thermoplastic matrix [36]. Ideally, the polymeric matrix totally masks the wood filler, avoiding contact between it and water. However, in this process, contact between wood filler and water occurred at the edge of the WPCs. This contact occurred particularly at the area of edge of the WPC samples. The water absorption rate may be very slow at the WPC surface but is higher on the edges of WPC samples as a result of preparation of samples for testing.

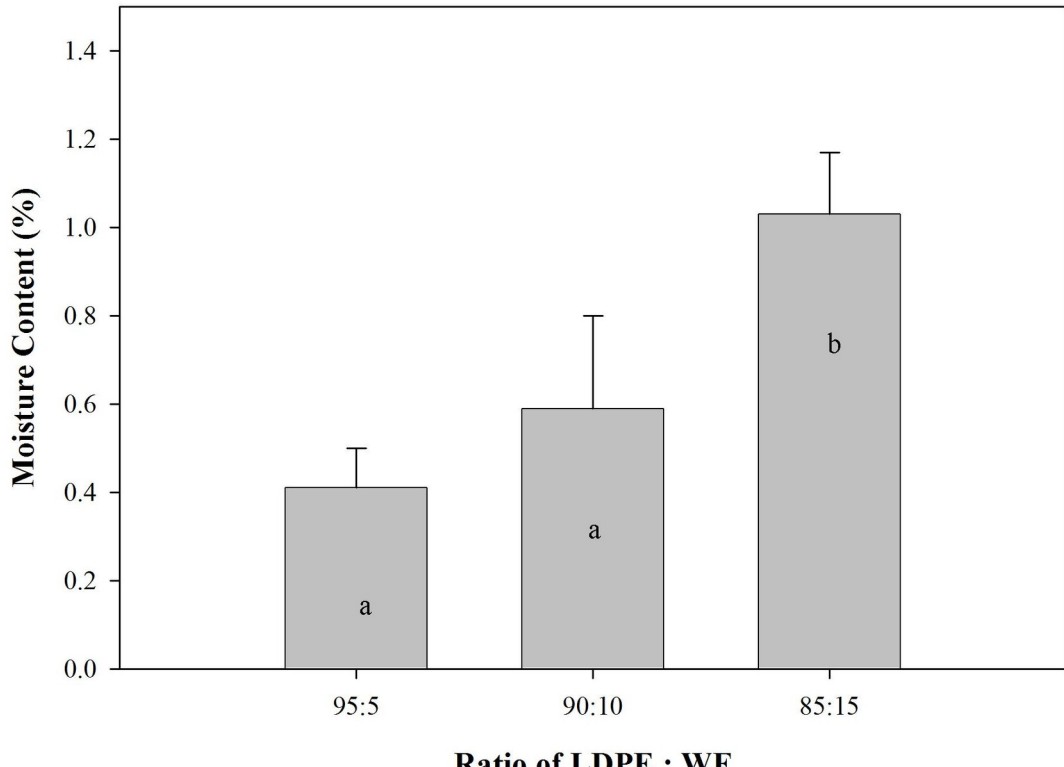

**Fig 3. Moisture content of predominantly plastic WPCs (Different letter at the bar are statistically different according to the DMRT at $\alpha = 0.05$).**

## Mechanical properties of the WPCs produced

Mechanical properties of the WPCs are shown in Figs 6 and 7, including modulus of elasticity (MOE), modulus of rupture (MOR), and internal bond (IB).

The MOE value of the WPC is presented in Fig 6. WPC systems showed higher MOE strength compared to the standard [25]. The MOE of WPCs usually fails to meet the standard because of the mismatch between the hydrophobic plastic matrix and the hydrophilic wood filler. Fortunately, in this case the extremely high proportions of plastic in the WPCs incorporated the wood filler without the aid of a coupling agent, thus the MOE was similar to that of LDPE plastic or even higher. The MOE strength of these WPCs was in the mean range of 3492–6059 N/mm$^2$ while MOE of LDPE alone is in the range of 1000–2000 N/mm$^2$ [37]. The values for MOE across the different WPCs showed the optimum ratio of LDPE to wood filler in the WPC was 90:10. Even though the MOE decreased when the proportion of wood filler was 15%, addition of WF up to 15% in the WPC system showed statistically non significant. Addition of wood filler to a WPC system without a compatibilizer will thus be optimal only up to a proportion of 10%.

Values of MOR for the WPCs were under the standard [25] and gradually decreased with increasing of wood filler content (Fig 7). Addition of wood filler in the WPC system at a proportion of up to 10% did not change the MOR strength as shown on DMRT results. MOR of neat LDPE is around 10 N/ mm$^2$ [38]. Strength was observed to change significantly at a proportion of 15% WF. This suggests increasing the proportion of wood filler in the WPC system causes poor MOR strength, making the WPC brittle. The MOR values were weak because of

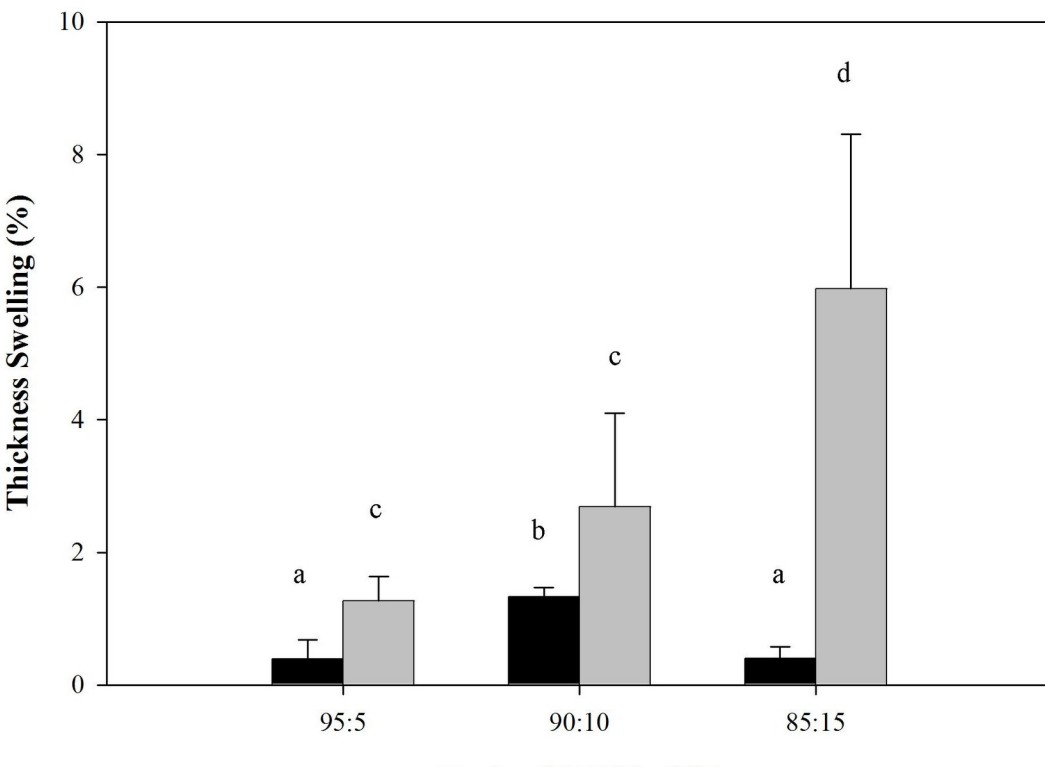

**Fig 4. Thickness swelling of predominantly plastic WPCs (Different letter upper the bars are statistically different according to the DMRT at α = 0.05).**

the inhomogeneity of the system [39]. The lack of uniformity in the dispersion of wood filler in the LDPE matrix resulted in weakness. With increasing proportions of wood filler, stress was not transferred and not distributed evenly across the WPC system.

IB testing, as depicted in Fig 8, showed very low values for IB. In addition, there is no significantly different in increasing WF on bonding quality of WPC. This indicated weak interfacial adhesion between the LDPE and the wood filler, resulting in low values for both MOR and IB. Research by Gao et al. [40] could explain this phenomenon. Many variables such as moisture content, type of wood particle used, and wood species used influence the final properties of any WPC. In Gao's study, the team observed more voids within the WPC system, many wood fibers pulling out of the matrix and much wood fiber breakage because of interactions between these factors. In the study presented here, the mixture of hydrophilic wood filler and hydrophobic LDPE without a coupling agent leads to similarly poor interfacial bonding.

## Degradation of the resulted WPC

The weight loss of the WPC samples after the graveyard test for defined observation periods is shown in Table 3.

The data presented in Table 3 show consistent weight losses for all treatment. Statistical analysis showed that condition of control was different significantly to the WPC sample. The longer the WPC sample was buried in the soil, the more weight loss occurred. Although plastics can hinder organisms attacking wood filler, the weight loss of these samples is challenging to explain. While wood can easily be degraded by soil organisms and insects, it is very difficult

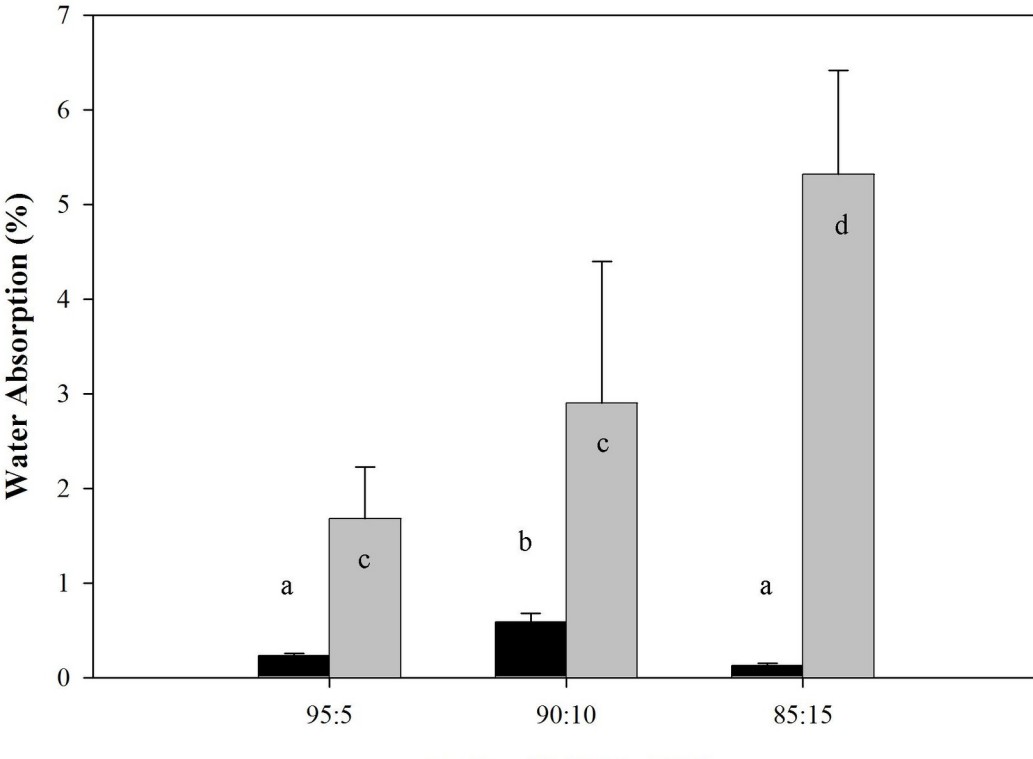

**Fig 5. Water absorption of predominantly plastic WPCs (Different letter upper the bars are statistically different according to the DMRT at α = 0.05).**

for these same agents to break up plastics. Therefore, there are factors present here that caused the WPC to deteriorate and these factors merit further investigation. Such factors may be determined to be either wood-decaying fungi or symbioses between moulds and termites degrading the wood filler.

The performances of the WPCs before and after being buried in the soil for 50 days are presented in Fig 9 while results of color analysis are presented in Table 4.

Macroscopically, color was altered both in tone and shade of WPCs. After being buried in the soil, both larger voids and increased number of voids were visible in the WPCs, particularly in the wood filler.

Interpretation of $\Delta L$ value prior to the graveyard test showed values of $<0$ or negative. This suggested the WPC color was dark. The darkness could be attributed to the extraction of the wood filler [41] but most influenced by heat treatment [42,43] which then influenced the final color of the WPC produced. The manufacturing process of the WPCs further involved high temperatures and solvents which can dissolve wood extractives [44] and penetrate into the WPC system, again influencing the color. After being buried in the soil, the color tended to positive which indicated the color of the WPC had lightened. This phenomenon of lightening color could originate from weathering factors such as relative humidity (RH) in the soil and lignin degradation of wood filler from the WPC. Durian wood has low class of durability [45] therefore extractive content within this wood was also very low or vanished. In other words, absence of extractive in durian wood made the resulted WPC dark and it was most influenced by heat treatment in process production of the WPC.

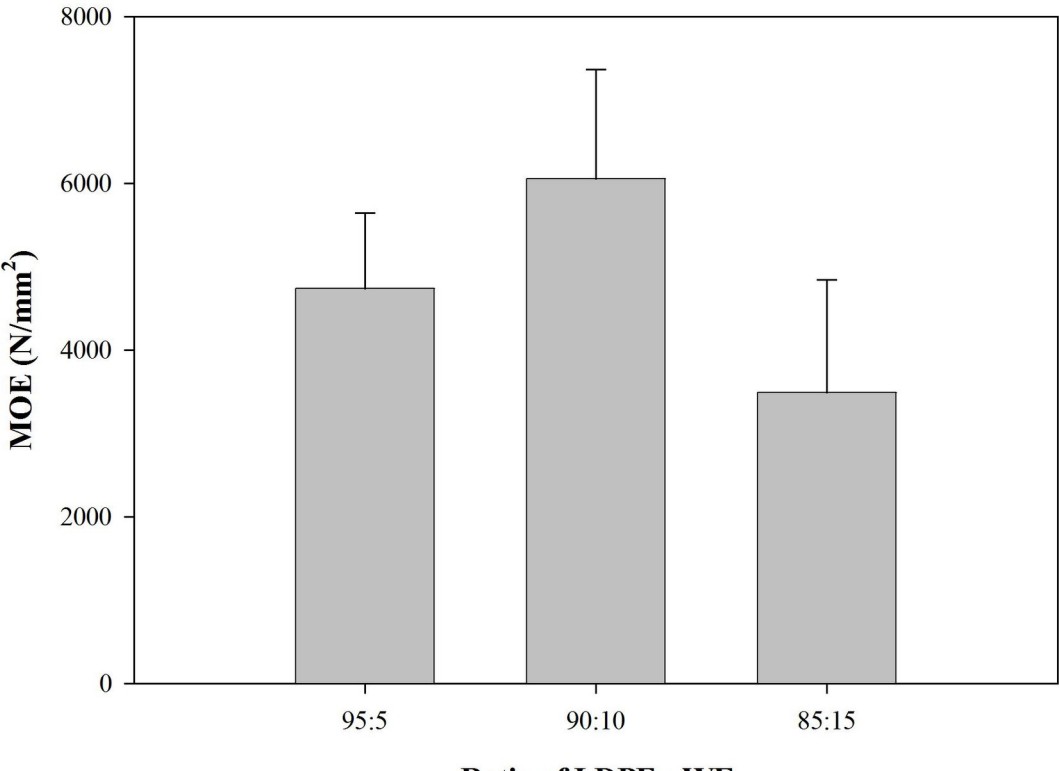

**Fig 6. MOE of predominantly plastic WPCs.**

In contrast, both the values of $\Delta a$ and $\Delta b$ were positive (+) prior to the graveyard test. Both values changed to negative (-) after the WPC was exposed to the soil. This change indicates that the WPCs' color altered from red into green and yellow into blue. This phenomenon suggests that soil factors such as moisture or relative humidity, high temperatures originating from exposure to sunlight, or possibly the activity of soil microorganisms were capable of degrading the WPC.

Overall values for color alteration ($\Delta E$) within both WPC 90:10 and WPC 85:15 were equal (43.3) while WPC 95:5 was low (25). This observation suggested that WPCs with a greater proportion of plastic up to 95% required more time to degrade, as indicated by discoloration. To confirm this degradation, ATR spectra of the WPC samples before and after being buried in the soil were evaluated. The results are depicted in Fig 10a and 10b.

When Fig 10(a)—before graveyard test—was compared to Fig 10(b)—after burial in the soil for 50 days—there was no alteration on the pure LDPE control sample. However, all specimens of WPC containing wood filler appeared to change spectra, particularly in the band of 500–1000 $cm^{-1}$ which could be assigned to detach the interface or interphase between wood and plastics.

The bands 700, 1500, 2800, and 2900 $cm^{-1}$ were quite similar to each other but different in the absorbance intensity of each chemical group. This difference suggested that alteration of chemical groups within the WPC sample had occurred. A strong peak at 1023 $cm^{-1}$ was related to wood spectra as a result of C-O stretching in cellulose and C-O deformation in lignin [46–48]. This peak suggested that wood was apparently present at the surface of the WPC sample after being buried in the soil. After the graveyard test, these ATR peaks at 1023 $cm^{-1}$ were

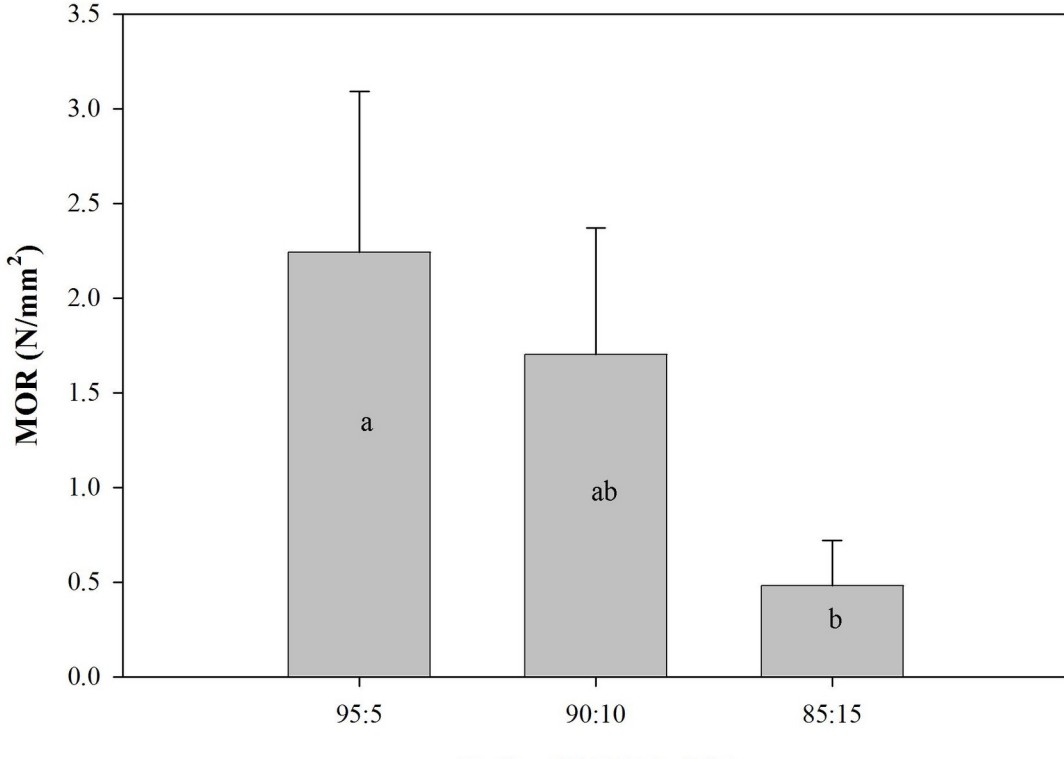

**Fig 7. MOR of predominantly plastic WPC (Different letter upper the bars are statistically different according to the DMRT at α = 0.05).**

greatly apparent for each of the WPCs (Fig 10b), indicating a loss of plastics masks and pull-out of wood component in WPCs system [49].

To evaluate the effect of temperature on the WPC composite system, DTA was employed. The results of DTA examination are presented in Table 5 which informed that addition of wood filler 5% and 10% made WPC products were more resistant to high temperature compare to either neat LDPE or WPC products with addition of 15% wood filler.

When the sample with the highest wood filler content was scanned, the temperature reported was under that of the LDPE sample (325˚C as compared to 390˚C for the LPDE control). This observation indicated that the addition of wood filler up to 15% decreased the quality of the WPC. Addition of wood filler between 5 to 10% increased the temperature. This phenomenon can be attributed to melting, decomposition, or alteration of the crystal structure [50] within the WPC system. This crystal structure is derived from either the LDPE thermoplastic or wood cellulose.

As termite bait, the WPCs lost weight continuously, even though the mass loss was very small, as shown in Table 6. Further, statistical analysis of Tukey test showed there was no different weight loss among the WPC samples.

Table 7 shows data on termite mortality for 7 days.

Termites feed on cellulose from wood and thus participate in decomposing wood [51]. In this study, the termites were feeding on WPCs and no other food was available. These termites only survived for 5 days with 100% mortality observed at day 6. Observation of termite guts was carried out to determine whether the termites had consumed both elements of the WPC

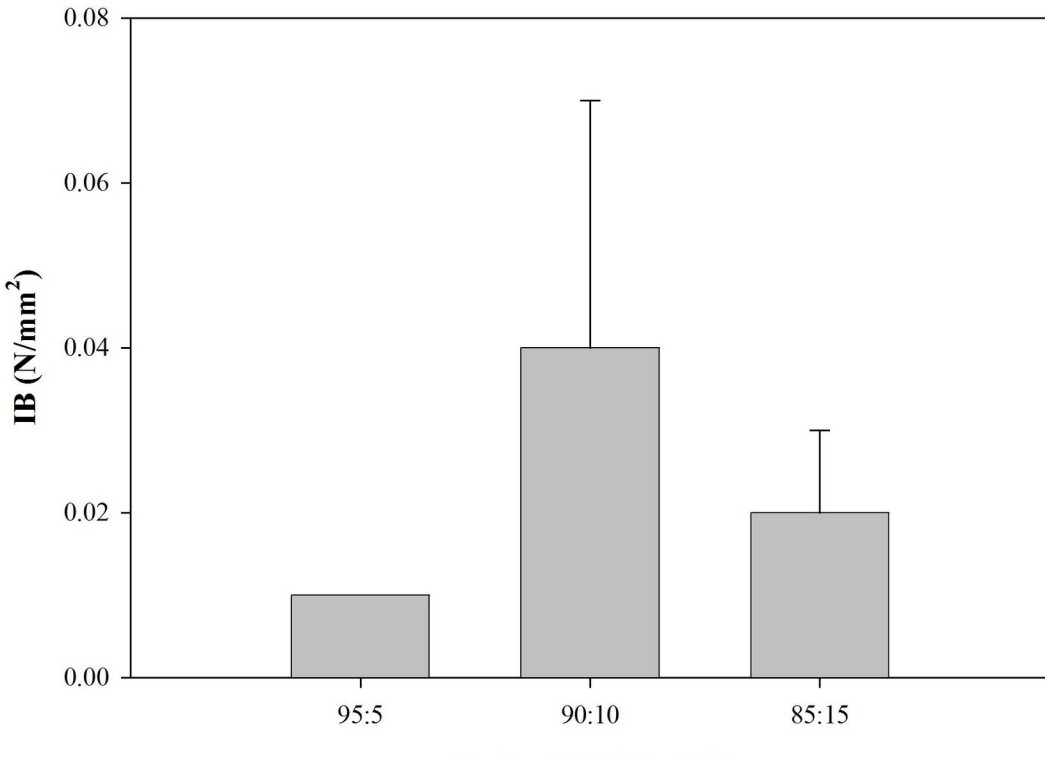

**Fig 8. IB of predominantly plastic WPC.**

or not. Light microscopy revealed alien objects in the termite guts, suspected to be microplastics, as shown in Fig 11.

Micro confocal Raman imaging spectrometry revealed that the termite guts contained small amounts of LDPE. The presence of LDPE indicates termites can consume plastics in the form of WPC even though they cannot digest the material. Successful identification of microplastics using this method has been reported by Karami et al. [52,53] in dried fish and commercial salts. In this study termite guts were prepared and observed either wet or dry and at both micro and macro resolution. Both the wet measurement process and the micro technique were unsuccessful (the spectra were not found) in identifying LDPE because of very low concentration of LDPE inside the termite guts. Fortunately, the dry measurement process and

**Table 3. Weight loss of the WPC after grave yard test in defined periods.**

| Type of sample<br>Weight loss (%) | WPC95:5 | WPC 90:10 | WPC 85:15 | Control |
|---|---|---|---|---|
| 10th day | 0.25 (0.01) | 0.63 (0.03) | 0.53 (0.09) | 8.66 (1.83) |
| 20th day | 0.63 (0.21) | 1.40 (0.12) | 1.37 (0.09) | 29.11 (7.51) |
| 30th day | 1.07 (0.15) | 2.12 (0.01) | 1.92 (0.01) | 43.13(8.73) |
| 40th day | 1.52 (0.26) | 2.72 (0.06) | 2.41 (0.01) | 58.64 (3.87) |
| 50th day | 2.23 (0.40)b | 3.53 (0.17)b | 3.03 (0.01)b | 76.23 (4.12)a |

Remarks: mean and standard deviation value in parentheses followed by different letter denotes that they are statistically not different according to the Tukey test at $\alpha$ = 0.05

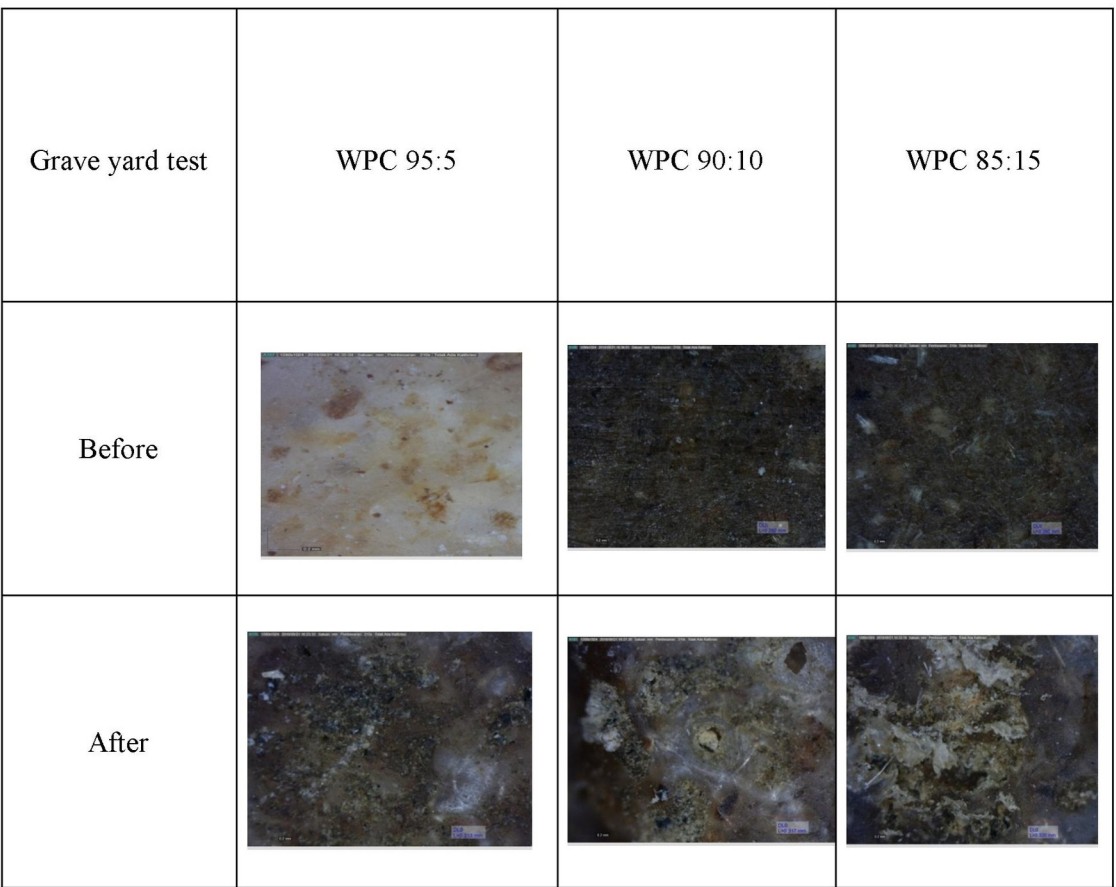

**Fig 9. Color performance of WPCs in various ratios of LDPE and wood filler.**

macro technique using a cuvette detect LDPE within termite guts in three spectra of 2732 cm$^{-1}$, 2863 cm$^{-1}$ and 2874 cm$^{-1}$, while the peak attributed to xylene may overlap in a spectra of 3054 cm$^{-1}$ as described in Fig 12.

## Conclusions

WPCs made of predominantly LDPE thermoplastic was successfully manufactured without a coupling agent by using the precipitation method. The physical properties of WPCs were

**Table 4. Results of color analysis before and after the 50-day graveyard test.**

| Parameter | Type of sample | WPC 95:5 | WPC 90:10 | WPC 85:15 |
|---|---|---|---|---|
| $\Delta L$ | Before | -30.0 | -45.1 | -45.4 |
|  | After | -9.3 | 2.1 | 1.1 |
| $\Delta a$ | Before | 11.6 | 6.9 | 8.0 |
|  | After | -2.8 | -1.0 | -2.4 |
| $\Delta b$ | Before | 16.7 | 6.3 | 7.0 |
|  | After | -5.5 | 1.6 | -2.0 |
| $\Delta E$ | Before | 36.2 | 46.1 | 46.6 |
|  | After | 11.2 | 2.8 | 3.3 |

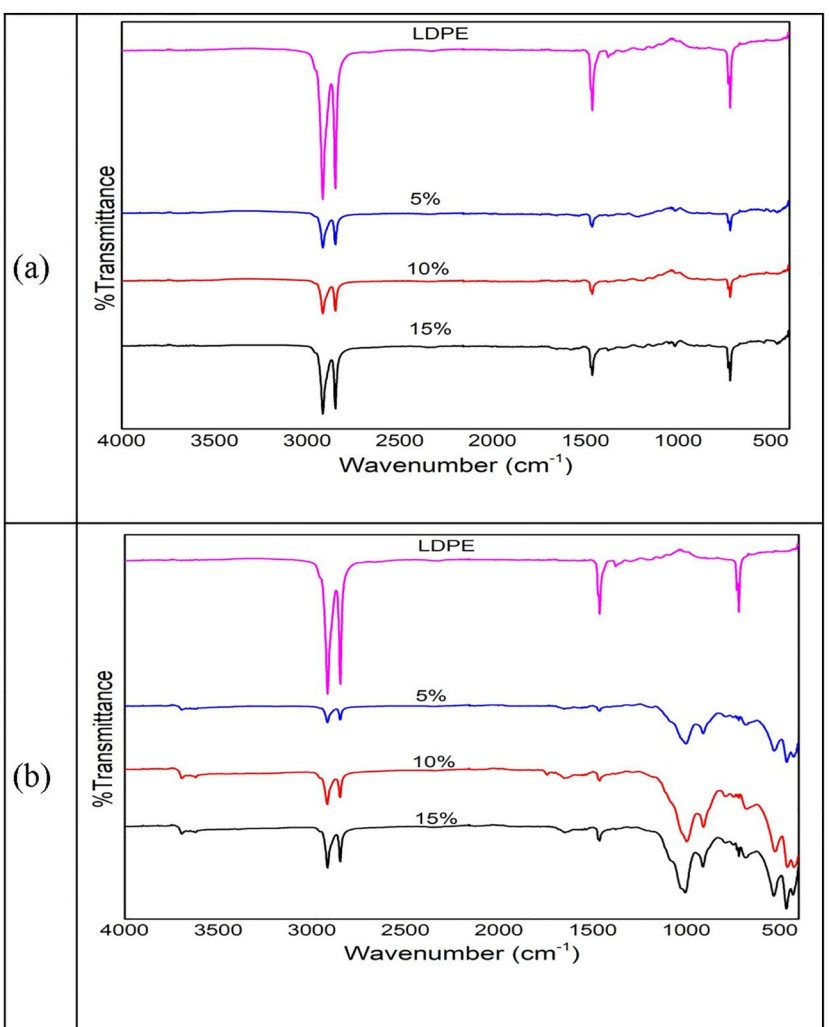

**Fig 10. ATR spectra of neat LDPE and the three WPCs before (a) and after being buried in the soil for 50 days (b).**

examined and demonstrated excellent characteristics as compared to those of conventional standard particleboard. Evaluation of WPCs mechanical properties produced a higher MOE but lower both MOR and IB. WPCs degradation was evaluated by the graveyard test and an assay test in termites. WPCs deteriorated in nature as shown by their discoloration after being buried in the soil and through a spectra alteration of ATR in the band of 500–1000 cm$^{-1}$ which could be assigned to detach the interphase between wood and plastics. As termite bait, the WPCs showed continual weight loss, even though the mass loss was very small. Detailed spectra of LDPE inside termite guts provide evidence that WPCs can be consumed by termites. This provides important evidence that plastics can be degraded by insects like termites in nature, though only broken up into microplastics, rather than completely biodegraded. Even

**Table 5. Results of DTA examination on LDPE and WPC with various compounds.**

| Type of sample | LDPE (Control) | WPC 95:5 | WPC 90:10 | WPC 85:15 |
|---|---|---|---|---|
| Temperature (˚C) | 390 | 435 | 405 | 325 |

**Table 6. Resistance of WPC from termite bait.**

| Type of WPC | Mean Weight Loss (%) | Class of resistance | Remarks |
|---|---|---|---|
| WPC 95:5 | 0.04a | I | Very durable |
| WPC 90:10 | 0.08a | I | Very durable |
| WPC 85: 15 | 0.06a | I | Very durable |

Remarks: mean followed by letter denotes that they are statistically not different according to the Tukey test at α = 0.05

**Table 7. Percentage of mortality for termites after assay test on WPCs for 1 week.**

| Treatment | Day | | | | | |
|---|---|---|---|---|---|---|
| | 1st | 2nd | 3rd | 4th | 5th | 6th |
| Without nest | 8 | 20 | 30 | 46 | 56 | 100 |
| With nest | 5 | 13 | 25 | 40 | 58 | 100 |

though the Raman spectra result was inconclusive as to the presence of LDPE was inside the termite guts, ATR spectroscopy, DTA, and color analysis demonstrated that WPCs can deteriorate when consumed by termites. This study thus combined physical, mechanical, chemical, and biological techniques to offer new knowledge on the mechanisms through which plastics can be modified with wood to produce new materials. In terms of concerns over the degradability of plastics in nature, it is likely that such WPCs—combination products made of predominantly waste plastics and small amount of wood—can be broken up with influence of heat (temperature), factors inside soils, and organisms like termites. The resulting microplastics, however, may remain an ecological concern in terms of their fate after digestion of the WPCs by termites.

(a) (b)

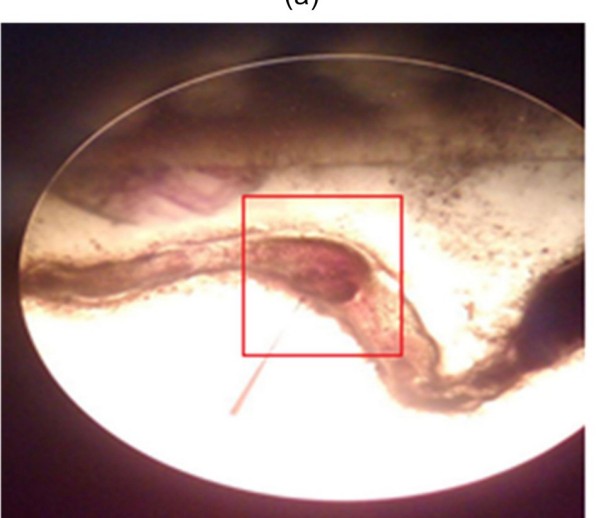 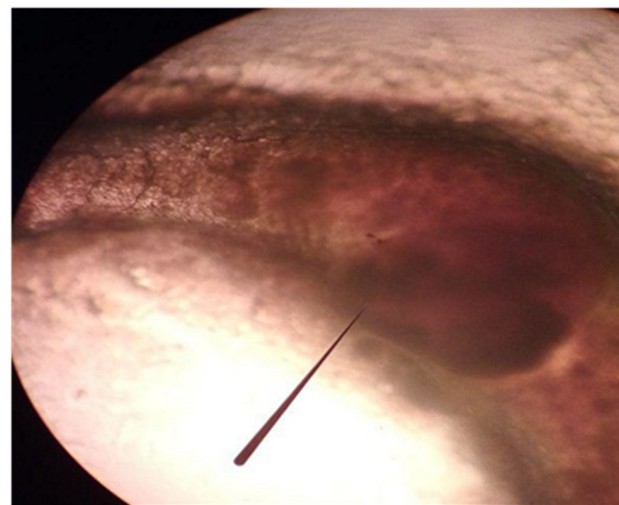

**Fig 11. An alien object in termite guts with 4x (a) and 10x magnifications (b).**

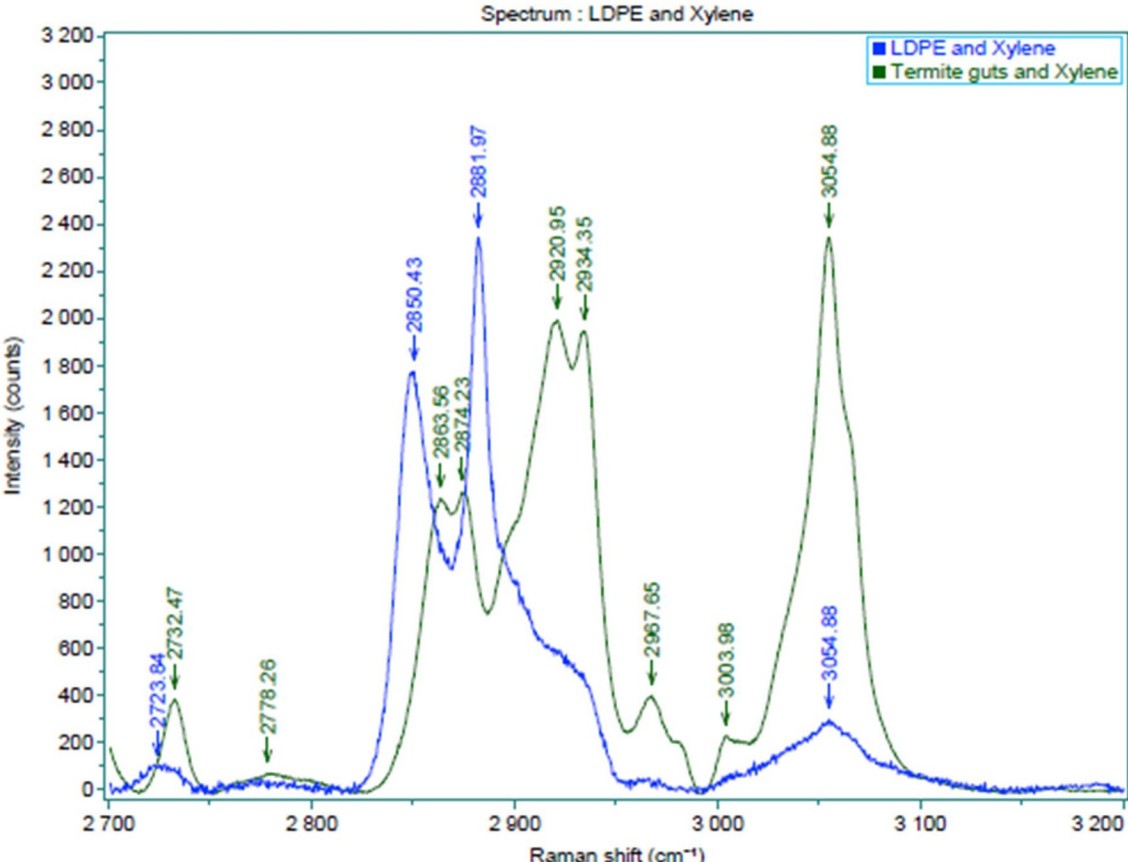

**Fig 12. Raman spectra suggesting LDPE inside termite guts (blue color indicates LDPE in xylene as reference; green color indicates termite guts dissolved in xylene).**

## Supporting information

**S1 Dataset.**
(DOCX)

**S2 Dataset.**
(DOCX)

## Acknowledgments

AN thanks to Dr. Emerson P. Sinulingga for making this collaboration research possible and to Dr. Delvian for their comments and critical reading of this manuscript.

## Author Contributions

**Conceptualization:** Arif Nuryawan, Iwan Risnasari, Rahmawaty.

**Formal analysis:** Arif Nuryawan, Nanang Masruchin, Ridwanti Batubara, Iwan Risnasari.

**Investigation:** Arif Nuryawan, Nova O. Hutauruk, Esra Yunita S. Purba, Fatih Khusno Satrio.

**Methodology:** Arif Nuryawan, Iwan Risnasari, Rahmawaty, Mohammad Basyuni.

**Project administration:** Arif Nuryawan, Nova O. Hutauruk, Esra Yunita S. Purba.

**Resources:** Nova O. Hutauruk, Nanang Masruchin, Fatih Khusno Satrio, Mohammad Basyuni.

**Supervision:** Arif Nuryawan, Ridwanti Batubara, Iwan Risnasari.

**Validation:** Arif Nuryawan, Nanang Masruchin, Deirdre McKay.

**Writing – original draft:** Arif Nuryawan, Nanang Masruchin, Deirdre McKay.

**Writing – review & editing:** Arif Nuryawan, Mohammad Basyuni, Deirdre McKay.

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
