## [Decision Letter · Decision Letter 0]

31 Mar 2020

PONE-D-19-35172

Properties of Wood Composite Plastics Made from Predominant Low Density Polyethylene (LDPE) Plastics and Their Degradability in Nature

PLOS ONE

Dear Dr Nuryawan,

Thank you for submitting your manuscript to PLOS ONE. After careful consideration, we feel that it has merit but does not fully meet PLOS ONE’s publication criteria as it currently stands. Therefore, we invite you to submit a revised version of the manuscript that addresses the points raised during the review process.

We would appreciate receiving your revised manuscript by May 15 2020 11:59PM. To enhance the reproducibility of your results, we recommend that if applicable you deposit your laboratory protocols in protocols.io, where a protocol can be assigned its own identifier (DOI) such that it can be cited independently in the future. For instructions see: http://journals.plos.org/plosone/s/submission-guidelines#loc-laboratory-protocols

We look forward to receiving your revised manuscript.

Kind regards,

Deniz Aydemir, PhD

Academic Editor

PLOS ONE

Journal Requirements:

2. Thank you for stating the following in the Competing Interests/Financial Disclosure* (delete as necessary) section:

"AN thanks to Universitas Sumatera Utara (USU) for funding this research under

scheme of Penelitian Terapan TALENTA-USU year of 2018, contract number

2590/UN5.1.R/PPM/2018 date of March 16, 2018."

We note that one or more of the authors are employed by a commercial company: "PT Horiba Indonesia, Jl"

Additional Editor Comments (if provided):

Dear Author,

Please edit all requested corrections and check the language of the manuscripts.

Best regards,

Reviewers' comments:

Reviewer's Responses to Questions

**Comments to the Author**

1. Is the manuscript technically sound, and do the data support the conclusions?

Reviewer #1: No

Reviewer #2: Yes

2. Has the statistical analysis been performed appropriately and rigorously? 

Reviewer #1: No

Reviewer #2: Yes

3. Have the authors made all data underlying the findings in their manuscript fully available?

Reviewer #1: No

Reviewer #2: Yes

4. Is the manuscript presented in an intelligible fashion and written in standard English?

Reviewer #1: No

Reviewer #2: No

5. Review Comments to the Author

Reviewer #1: While variation of WF content is linear, variation of properties is not linear; this aspect is not discussed. Authors marginally discussed about extractives from wood dissolved in solvent and then dispersed in PE matrix. Therefore the final WPC material consists of PE matrix, dispersed WF particles and dispersed extractives from WPC. Actually the presence of extractives dispersed in PE is the key factor in this system, since this strongly affect the properties of final material. Therefore, a complete study should consist of four types of samples: 1) PE alone; 2) PE and extractives only (WF in xylene until complete extraction – residual WF in new xylene shows no coloration – , filtration, PE in xylene containing extractives); 3) PE and residual WF (PE and WF remained after complete extraction from sample 2; 4) PE and WF (samples presented in this paper). Analysis of extractives in xylene is needed (e.g. FTIR). Careful language check is absolutely needed.

- Materials: sieved sawdust is wood particulates, not wood fibers.

- Dissolution/re-precipitation technique: Dissolution temperature (115 oC) is close to boiling point of xylene (144 oC); how was evaporation of xylene avoided, in order to maintain the 1:20 ratio? What was the mixing time after addition of WF? How was the liquid mixture transformed into pellets?

- Composite processing: Weight of 15 kg is for entire surface of the board? This is much lower than 30 kgf/cm2 for hot press. Why is cold press used after hot press?

- Physical properties: How many samples have been tested in order to perform statistical analysis? Moisture content depends on the storage conditions, since just after preparing moisture content is virtually zero. Therefore, without describing storage conditions, moisture content is meaningless. Considering the strong hydrophobicity of PE, major component, swelling, if any, might occur only at the margins of the specimen, therefore dimensional change will manifest mainly at the margins, and less inside; was this considered? Thickness of specimens was not mentioned. Figure 1: Why 90:10 mixture had highest density? Agglomeration of WF, as described by authors, is expected since no compatibilizer was used. Final material has low homogeneity thus large variation in properties, which can induce misleading interpretation of results, and made them unsuitable for practical purpose. What means a, b, c, d in Figure 2 and Figure 3? Why 90:10 mixture had highest swelling and water absorption at 2 hours since it also had highest density?

- Mechanical testing: Size of specimens or standard used is not mentioned. Without using PE alone as references, discussion on addition of WF is meaningless. Replace OF with OR in “Bending strength of modulus of rupture”. Which parameter is “MOE strength”? Should be either MOE, either strength. Delete “dimensions” from were determined in “mm dimensions”. Please introduce values of MOE and MOR for neat LDPE in Figures 5 and 6. What type of device has been used for internal bond (IB) testing? Please specify if bending tests were performed on vertical or horizontal position of the samples.

- Degradation: Without describing the exposure condition (e.g. sample size/mass, soil temperature, humidity, pH, graveyard test is meaningless). Since adding WF could increase degradability of plastics, PE alone should be considered as reference, for all tests, as in Figure 9. How many samples have been buried in the soil for degradation testing?What means the values in brackets in Table 3? Why the 90:10 mixture had highest mass loss? Please provide a reference for equation 8 and upgrade the reference Hunter Lab. 2008 (is just direct link to Hunter Lab webpage) – e.g. https://support.hunterlab.com/hc/en-us/article_attachments/201439655/an07_96a.pdf

Table 4: It is not clear what is the reference sample in color analysis. Change in colour could be only an effect of leaching the extractives into soil, not of degradation of WF itself. From all information that DTA analysis can provide, it is not clear what is the meaning of temperature in Table 5. The values are by far over the alteration of crystalline structure of PE or melting. They are in the range of thermal degradation, which is a complex process, especially in the case of WF, therefore careful interpretation is needed. DTA curves should be given. Figure 11: Normaly, preparation of WPC should assure that xylene solvent is totally removed from the system. Replace ENCAPSULATED with INCORPORATED in “…termites have no choice in feed sources except from wood sawdust particles encapsulated in the matrix plastics in WPC systems. Authors mentioned “acclimatization to the termite was conducted for 10 days after collecting termites from the native habitat.”, but how long were the termites in contact with WPC samples?

Reviewer #2: Dear Editor,

The paper is related to “Properties of Wood Composite Plastics Made from Predominant Low Density Polyethylene (LDPE) Plastics and Their Degradability in Nature”. The paper fits within the scope of the journal. I think the contribution of the data in the article to the research community will be limited. The study has to go under major language sub editing. Work can be published after language editing.

Best regards.

Assistant Professor Dr.Timucin BARDAK

Bartin University

Furniture and Decoration Program, Bartin Vocational School,

74000, Bartin - Turkey

E-mail: timucinb@bartin.edu.tr

6. PLOS authors have the option to publish the peer review history of their article (what does this mean?). If published, this will include your full peer review and any attached files.

Reviewer #1: No

Reviewer #2: No

---

## [Author Response · Author response to Decision Letter 0]

23 Jun 2020

Thank you for organizing the review of our manuscript titled as ‘Properties of Wood Composite Plastics Made from Predominant Low Density Polyethylene (LDPE) and Their Degradability in Nature’ (ID: PONE-D-19-35172) submitted to Plos One.

This manuscript is one of research results of our group particularly on behalf of the Center Excellence for Mangrove of Universitas Sumatera Utara (USU), and it is also funded by USU. 

In response to the journal and reviewers’ comments, we have added further detail on supporting information as well as experimental process, as requested. For instances details of statistical analysis, discussions of linearity of addition of wood filler into wood composite plastics (WPC) system, technical production of WPC, technical testing of mechanical properties, exposure condition for degradation examination including termite assay test. These revisions appeared and incorporated in the resubmit manuscript in blue and green color as well as additional documents with answers point by point to editor and reviewer’s comments.

Authors of this manuscript have declared that there is no competing interest because most of them work under USU institution. One of author has had an affiliation with a commercial company that provides technical services to USU laboratory. Dr. McKay is employed by Keele University and we have a memorandum of understanding (MoU) facilitating shared research between the two institutions, covering several shared workshops, co-supervision and joint publication. There is thus no conflict of interest to report.

I am sure that aforementioned information will help better understand of this manuscript. The authors hope that the manuscript has been amended suitably for publication in "Plos One" and that you are happy with the current revised manuscript. Please let me know if you need further information on the manuscript. My mailing address, e-mail address, phone and fax numbers are given below:

Current mailing address

Department of Forest Products Technology

Faculty of Forestry,

Universitas Sumatera Utara

Jl. Tri Dharma Ujung No.1 Kampus USU

Medan, North Sumatra, Indonesia 20155

Tel: +62-877-6918-7088

Fax: +62-61-820-1920

E-mail: arif5@usu.ac.id

Best regards,

Arif Nuryawan, Ph.D

---

## [Editor Report · Decision Letter 1]

8 Jul 2020

Properties of Wood Composite Plastics Made from Predominant Low Density Polyethylene (LDPE) Plastics and Their Degradability in Nat ure

PONE-D-19-35172R1

Dear Dr. Nuryawan,

We’re pleased to inform you that your manuscript has been judged scientifically suitable for publication and will be formally accepted for publication once it meets all outstanding technical requirements.

Kind regards,

Deniz Aydemir, PhD

Academic Editor

PLOS ONE
---

## [Editor Report · Acceptance letter]

16 Jul 2020

PONE-D-19-35172R1 

Properties of wood composite plastics made from predominant low density polyethylene (LDPE) plastics and their degradability in nature 

Dear Dr. Nuryawan:

I'm pleased to inform you that your manuscript has been deemed suitable for publication in PLOS ONE. Congratulations! Your manuscript is now with our production department. 

Kind regards, 

on behalf of

Dr. Deniz Aydemir 

Academic Editor

PLOS ONE